# Heat Transfer Enhancement of Indirect Heat Transfer Reactors for Ca(OH)₂/CaO Thermochemical Energy Storage System

**Boyan Wang** [1,2] , **Zhiyuan Wang** [1,2,*] , **Yan Ma** [1,2] **and Yijing Liang** [1,2]

1 School of Energy and Power Engineering, University of Shanghai for Science and Technology, Shanghai 200093, China; wangboyan1996@163.com (B.W.); my18395588645@163.com (Y.M.); 203630208@st.usst.edu.cn (Y.L.)
2 Shanghai Key Laboratory of Multiphase Flow and Heat Transfer in Power Engineering, Shanghai 200093, China
* Correspondence: wangzhiyuan@usst.edu.cn; Tel.: +86-139-0160-0456

**Abstract:** The efficiency of a thermochemical energy storage system can be improved by optimizing the structure of the thermochemical energy storage reactor. We proposed two modified structures for indirect heat transfer thermochemical energy storage reactors for a Ca(OH)₂/CaO system to improve their heat transfer performance. Our results showed that improving convective heat transfer offered varying effects on heat transfer performance in different reaction processes. For a half-plate pin fin sinks (HPPFHS) reactor and a plate pin fin sinks (PPFHS) reactor, enhancing the convective heat transfer process could improve the heat transfer performance in the dehydration process for a porosity of 0.5, and the time needed to complete reaction was reduced by around 33% compared with plate fin sinks (PFHS) reactor. As for the hydration process, because heat conduction along the bed dominated heat transfer performance, this method had little effect. Furthermore, we found that enhancing heat conduction along the bed and convective heat transfer had different effects on reaction process at different reaction areas. The HPPFHS reactor had a lower pressure drop along the HTF channel and exorbitant velocity of heat transfer fluid (HTF) was unnecessary. Under the condition of the bed porosity of 0.8, due to the lower thermal conductivity of material, both modified reactor structures had little effect on dehydration. However, because the temperature difference between bed and HFT was bigger, the PPFHS reactor could reduce the time of completing the hydration reaction by 20%. Above all, when planning to modify the reactor structure to improve the heat transfer performance to enhance the reaction process, the heat conditions along the bed, convective heat transfer between HTF and the bed and material parameters should be considered totally.

**Keywords:** thermochemical energy storage; heat transfer enhancement; Ca(OH)₂/CaO



## 1. Introduction

Acquiring, using and storing energy is extremely important as energy demand has been increasing recently. As we face the gradual depletion of fossil fuels and the pollution caused by their usage [1], it is urgent to replace fossil energy with a clean and renewable energy source. Solar energy is widely used to generate electricity due to its wide distribution, large reserves and mature technology. However, solar power generation is greatly affected by seasons, weather and many other factors which could make energy supply unstable [2]. Energy storage systems applied in the field of solar power generation can regulate the electricity network pressure by storing heat when the electricity consumption is at a low ebb and release it to generate electricity when it is at its peak [3]. Current energy storage technologies applied to power networks includes pumped hydropower, compressed air, batteries and thermal energy storage. Thermal energy storage systems could be divided into latent energy storage, sensible energy storage and thermochemical energy storage (TCES) systems [4,5]. Among them, TCES offers more energy storage cycles, longer transportation distance and higher energy storage density. Slaked lime(Ca(OH)₂)/quicklime(CaO) pair is

a kind of relatively mature TCES system the operating temperature of which is between 623.15 K and 723.15 K in general. In the charging step, the heat is transferred by the heat transfer fluid (HTF) from solar power tower to the TCES reactor to trigger the endothermic decomposing of slaked lime into quicklime and water. During the discharging step, the heat generated from the exothermic hydration of quicklime and water, is transferred by the HTF to the power station for electricity generating set. Steam is used as a reactant, which is reacted with CaO during the hydration process, and is generated by the steam generator and conveyed into the reactor bed by a compressor [6]. Depending on the structure of the reactor, air or steam is used as HTF, which will be later discussed. Heat is released and stored by the following reaction in a CaO/Ca(OH)$_2$ system:

$$Ca(OH)_2 \leftrightarrow CaO + H_2O \; \Delta H_{forward} = 104 \; kJ/mol$$

The forward reaction (charging process) is endothermic, and the reverse (discharging process) is exothermic. In the redox reaction system, the reactants need to be heated to activate the reaction. Reactors for such systems can be classified as indirect heat transfer reactors and direct heat transfer reactors. Under direct heat transfer reactor, the heat is carried directly into the porous media bed by the steam flow, and the steam can be used as HTF and reactant [7]. For the indirect heat transfer reactor, the HTF and steam are transported in a crossflow scheme. The HTF (Air) is used to transfer the heat in the system and steam is used as reactant [8]. In the indirect heat transfer reactor, there were two types of heating patterns, as shown in Figure 1: tubes and plate fin sinks (PFHS) reactor. The Ca(OH)$_2$/CaO TCES system has been studied extensively in the fields of reaction kinetics and application. Linder et al. [9] investigated the reaction kinetics and mathematical model for a 10-kW reactor containing 20 kg of reaction materials and analyzed the heat loss of the reactor. Schäuble et al. [10] explored the effect of the H$_2$O partial pressure on the reaction and developed a complete dynamic model to simulate the reaction process in the reactor. Additionally, the heat and mass transfer along the similar porous bed was also researched [11]. Basing on the above works, Qasim et al. [12] made a mature numerical model about indirect heat transfer reactor for CaO/Ca(OH)$_2$ TCES system.

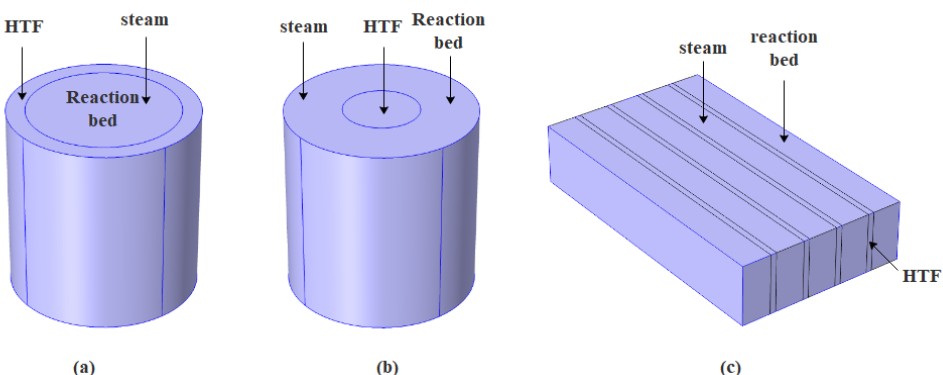

**Figure 1.** Indirect heat exchange structure of the Ca(OH)$_2$/CaO thermochemical energy storage reactor. (**a**) External flow of the tubular reactor. (**b**) Internal flow of the tubular reactor. (**c**) Plate fin heat sinks (PFHS) reactor.

In case of the indirect heat transfer reactor, heat absorbed and released from the redox reactions is transferred by HTF, and the heat transfer process is mainly composed of the heat conduction in the bed and the convective heat transfer between HTF and bed. As the reactions are closely related to the temperature in the bed, the heat transfer performance of the reactor could affect the reaction process and thermal efficiency of the reactor. However, the heat conductivity of Ca(OH)$_2$/CaO is very low, which reduces the efficiency of the reactor greatly. To overcome this problem, two methods are adopted at present. In the first method, direct improvement of thermal conductivity of materials

can fundamentally solve this problem. For example, Funayama et al. [13] used ceramic honeycomb to support the material to directly improve its thermal conductivity. In case of the second method, improving heat transfer performance of the reactor is feasible by modifying the heat transfer structure of the reactor. For example, Qasim et al. [14] studied the heat transfer performance of the tube-type reactor by adopting heat-conducting fins, which improved the heat conduction along the bed. Chen et al. [15] also took a tube-type indirect heat transfer reactor as the research object to discuss the strengthening effect of enhancing the heat conduction along the bed on the reaction process. However, there was still little relevant research for plate-type indirect heat transfer reactors and the effect of enhancing convective heat transfer performance on the reaction process. Considering that the heat transfer process in the plate-type indirect heat transfer reactor is similar to that in the plate fin sink, in this work we modify the structure of the TCES reactor to enhance heat transfer performance of the reactor based on the existing related research on plate fin heat sinks.

In the field of heat transfer enhancement of plate fin heat sinks, adding pin fins into the flow channel is an effective and mature method to enhance its convective heat transfer performance. This method can disturb fluid flow to destroy the temperature boundary layer and increase the velocity of fluid near the fins, through which the convective heat transfer performance is improved. A lot of research applied this technology. Yu et al. [16], Yang et al. [17] and Yuan et al. [18] added pin fins into the heat sink and located them in the middle of the flow channel to enhance heat transfer performance. They explored the changes in the thermohydraulic and heat transfer performance by analyzing the Nusselt number, thermal resistance and pressure drop after adopting the pin fins. The Nusselt number and thermal resistance described the intensity of convective heat transfer, and the pressure drop revealed the power consumed by an air compressor to maintain fluid flow. The results showed that adding pin fins improved the heat transfer performance of the radiator with the increased pressure drop. Furthermore, some researchers proposed that the permutation and trait of pin fins also had influences on heat transfer performance. Wang et al. [19] found that the half-plate pin fin heat sinks (HPPFHS) had a better performance than the plate fin heat sink. Freegah et al. [20] further studied the influence of the permutation of half-round fins on the heat transfer performance. They explored the heat transfer performance of the horizontally and vertically arranged symmetrical and corrugated half-round pins used for a plate heat sink. The experimental results revealed that the vertical arrangement of half-round pins relative to the flow channel used for plate heat sinks had a higher Nusselt number, and lower thermal resistance and base temperature than those with a horizontal arrangement. The symmetrical arrangement of pins also had the same law when compared with a corrugated arrangement. In summary, symmetrical half-round pins in vertical arrangement used for plate heat sink exhibited the best reinforcing effect. These studies analyzed the changes in the Nusselt number and thermal resistance to explain the intensity of heat transfer.

Overall, we noticed that the main method of the heat transfer enhancement in $Ca(OH)_2/CaO$ thermochemical energy storage reactor was applying heat conduction sheets in the bed, by which only the heat conduction is strengthened. However, the reports about the enhancement of convective heat transfer between HTF and bed are still rare. In this paper, we tried to use the method of heat transfer enhancement of plate fin heat sinks to enhance the TCES reaction process in the indirect heat transfer reactor. Two modified structures of the indirect heat transfer reactors were proposed, and their effects on the heat conduction, convective heat transfer and reaction process were compared. The specific structure of the reactor will be described in detail in Section 2. Finally, we summarized some problems that should be focused on in the heat transfer enhancement of thermochemical energy storage reactor for $Ca(OH)_2/CaO$ system. We hope that this work could provide some useful information about the further application of thermochemical energy storage technology.

## 2. Mathematical Model

### 2.1. Reactor Geometry and Hypothesis

Based on the plate-type indirect heat transfer reactor, two kinds of new structures were proposed, as shown in Figure 2. The first modified reactor was a half-plate pin fin heat sinks (HPPFHS) reactor whose half-round fins were arranged symmetrically and vertically. The other was a plate pin fin heat sinks (PPFHS) reactor whose fins were placed in the middle of the flow channel vertically. The channel width of the HTF channel was 3 mm, and the radius of the cylindrical fin was 0.5 mm. The first fin was located at 5 mm along the flow direction, the distance between each fin is 20 mm and a total of 10 groups of fins were set. To reduce the computation, we simplified the three-dimensional fixed-bed model (Figure 1) to a two-dimensional model by considering steam flow direction the same as the HTF flow direction (Figure 3). In order to suit the boundary condition selected based on the relevant experiment data, the length of the reactor was limited to 200 mm [21].

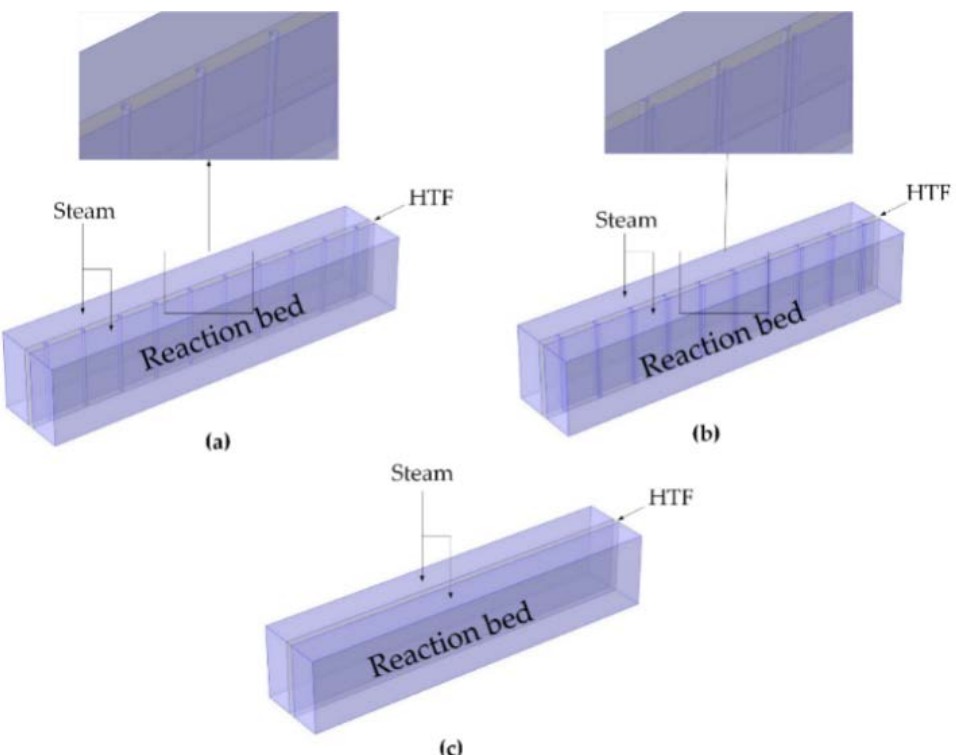

**Figure 2.** One research unit of the reactor. (**a**) Plate pin fin heat sinks (PPFHS). (**b**) Half-plate pin fin heat sinks (HPPFHS). (**c**) Plate fin heat sinks (PFHS).

We used the k–ε turbulence model to deal with the vortex created by fins in PPFHS and HPPFHS reactors [16–20]. The specific size of the reactor and relevant simulation parameters is given by Table 1. The boundary conditions were set up based on the environmental conditions (0 K, 1 atm.). The following assumptions were applied to this model:

- The porous bed was treated as a continuum, and the reaction bed porosity remained constant in the dehydration/hydration process;
- The effective thermal conductivity was constant;
- The density of the reactant solid changed with the conversion of the reactant;
- The specific heat at constant pressure changed with temperature;
- The fluid flow of HTF was the two-dimensional steady state flow at different temperature and pressures.

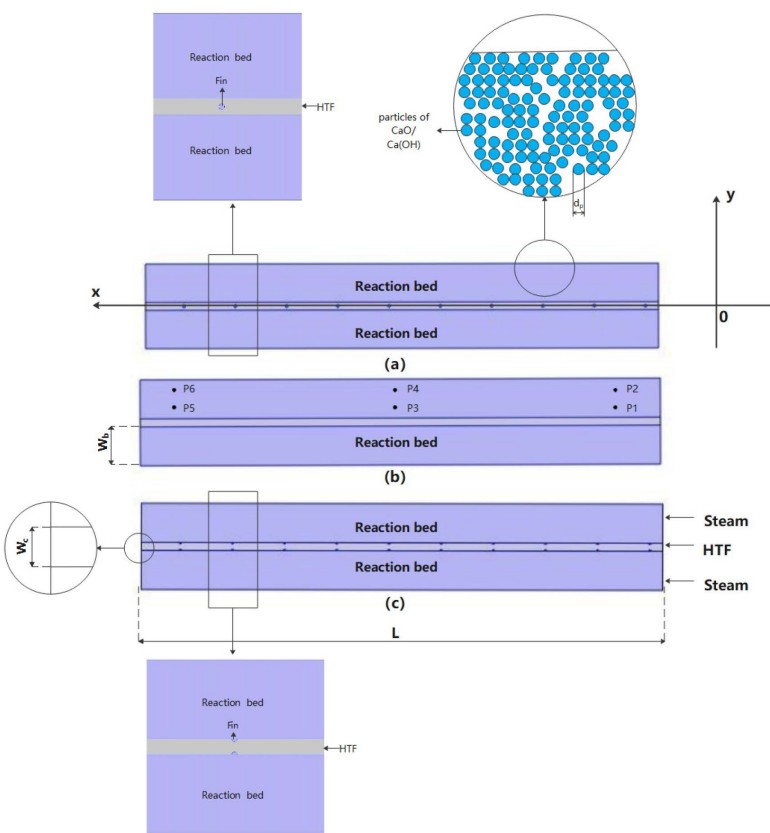

**Figure 3.** Two-dimensional models of three kinds of heat exchange structures for TCES reactors. (**a**) Plate pin fin heat sinks reactor. (**b**) Plate fin heat sinks reactor. (**c**) Half-plate pin fin heat sinks reactor.

**Table 1.** Geometric dimensions and reaction parameters.

| Parameter | Symbol | Value |
|---|---|---|
| Width of porous bed | $W_b$ | 15 (mm) |
| Width of flow channel | $W_c$ | 3 (mm) |
| Length of porous bed | L | 200 (mm) |
| Density of CaO [7,18] | $\rho_{CaO}$ | 1.666 (g/cm$^3$) |
| Density of Ca(OH)$_2$ [7,18] | $\rho_{Ca(OH)2}$ | 2.200 (g/cm$^3$) |
| Reaction enthalpy | $\Delta H$ | 104,000 (J/mol) |
| Pre-exponential factor dehydration [7] | $A_d$ | $715 \times 10^7$ (1/s) |
| Activation energy of dehydration [7] | $E_d$ | $187 \times 10^3$ (J/mol) |
| Pre-exponential factor hydration [7] | $A_h$ | $53 \times 10^3$ (1/s) |
| Activation energy of hydration [7] | $E_h$ | $83 \times 10^3$ (J/mol) |
| Effective thermal conductivity of reactant solid [18] | $\lambda_{eff}$ | 0.4, 0.1 (W/m·K) |
| Velocity of air at entrance | u | 25, 30, 35 (m/s) |
| Porosity [18] | $\varepsilon$ | 0.5, 0.8 |
| Radius of cylindrical fins | r | 0.5 (mm) |
| Diameter of grain [6] | $d_p$ | 5 (μm) |

### 2.2. Mathematical Model

The specific reactor mathematical model was based on a previous study [12]. The governing equations of mass, momentum, turbulent kinetic energy, turbulent energy dissipation rate and energy in the steady turbulent flow of HTF were solved by the standard k–ε model.

Energy governing equation in porous bed is as follows:

$$(\rho C)_{eff} \cdot \partial T_{bed}/\partial t + \rho_{st} \cdot C_{st} \cdot u_{st} \cdot \nabla T_{bed} + \nabla \cdot (-\lambda_{eff} \cdot \nabla T_{bed}) \pm \dot{S}_Q = 0$$

$$\dot{S}_Q = (1 - \varepsilon) \cdot \Delta H \cdot \dot{R} \quad \dot{R} = V_{rs} \cdot dX/dt$$

where $\dot{S}_Q$ is the reaction heat source which is generated by the endothermal or exothermic reactions of the material, and $\Delta H$ is the reaction enthalpy. $V_{rs}$ is the molar density of the remaining reactant, and $(\rho C)_{eff}$ and $\lambda_{eff}$ are effective thermal conductivity and density, which depend on the porosity of the bed as follows:

$$(\rho C)_{eff} = (1 - \varepsilon) \cdot (\rho C)_{bed} + \varepsilon \cdot (\rho C)_{st}$$

$$\lambda_{eff} = (1 - \varepsilon) \lambda_{bed} + \varepsilon \lambda_{st} \lambda_{eff} = (1 - \varepsilon) \lambda_{bed} + \varepsilon \lambda_{st}$$

where $(\rho C)_{bed}$ is the bed heat capacity that changes with the conversion (X) of the solid reactant and the specific heat of the solid reactant at a constant pressure changes with temperature of the bed:

$$(\rho C)_{bed} = (1 - X) \cdot (\rho C)_{CaO} + X \cdot (\rho C)_{Ca(OH)_2} \quad C_{CaO} = 0.1634 \frac{J}{kg \cdot K^2} \cdot T_{bed} + 844 \frac{J}{kg \cdot K}$$

$$C_{Ca(OH)_2} = 0.3829 \, J/\left(kg \cdot K^2\right) \cdot T_{bed} + 1323.4 \, J/(kg \cdot K)$$

The mass equation of the water vapour transferred in the porous media bed can be described as follows:

$$\partial(\varepsilon \rho_{st})/\partial t + \nabla(\rho_{st} + u_{st}) \pm \dot{S}_m = 0$$

$\dot{S}_m = (1 - \varepsilon) \cdot \dot{R} M_{st}$ where $\varepsilon$, $\rho_{st}$ and $u_{st}$ are the porosity, density and velocity of the saturated steam. $S_m$ is the change in steam mass caused by the chemical reaction, R is the reaction rate and $M_{st}$ is the molar mass of the steam. Darcy's law was used to describe the steam flow in porous beds, and the permeability model based on the Kozeny–Carman equation is as follows [8]:

$$u_{st} = -K/\eta_{st} \cdot \nabla P_{st} \quad K = \left(d_p^2 \cdot \varepsilon\right)/\left(\psi(1 - \varepsilon)^2\right)$$

where $P_{st}$ is the partial pressure of the steam in the bed, K is the bed permeability and $\eta_{st}$ is the viscosity of the steam. $d_p$ is the diameter of $CaO/Ca(OH)_2$ particles, and it was taken as 5 µm in the work. $\psi$ is the Kozeny–Carman factor, which was 180 [8].

The conversion (X) of solid reactant expressed by the empirical model of the gas-solid reaction is as follows [22]:

$$dX/dt = K(T)f(X)h(P_{st}, P_{eq})$$

where f(X) is described by the first-order model (F1):

$$f(X) = -(1 - X).$$

Ogura et al. [8] conducted experiments for partial pressure of steam ($P_{st}$) and the pressure at equilibrium ($P_{eq}$) Based on this result, the pressure term can be settled as:

$$h(P_{st}, P_{eq}) = (1 - P_{st}/P_{eq})^n, n = 1.$$

K(T) is the reaction rate constant described by the Arrhenius equation [23]:

$$K(T) = A \cdot \exp((-E)/RT).$$

Because of the relationship between $P_{st}$ and $T_{eq}$ (the temperature at equilibrium), the pressure term can be shown as the temperature [7]:

$$\ln\left(P_{bed}/10^5\right) = -12,845/T_{eq} + 16.508.$$

Therefore, the first derivative of conversion with respect to time is

$$dX/dt = \mp(1-X) \cdot A \cdot \exp((-E)/(RT_{bed})) \cdot (T_{bed}/T_{eq} - 1)$$

where $T_{bed}$ is the temperature of the porous bed. Initial and boundary conditions used for the energy storage and release steps are listed in Table 2 with respect to the coordinates indicated in Figure 3. The inlet velocity boundary was used in this model so that the pressure (P) at the inlet of the flow channel was used to assess the change in pressure drop after adding the fins.

**Table 2.** Boundary/initial conditions for governing equations.

| Boundary/Initial Conditions | Description |
|---|---|
| $T_{bed}(x,y,t=0) = T_{HTF}(x,y,t=0) = 623.15$ K | Initial temperature of hydration |
| $T_{bed}(x,y,t=0) = T_{HTF}(x,y,t=0) = 723.15$ K | Initial temperature of dehydration |
| $T_{in/D} = 863.15$ K, $T_{in/H} = 623.15$ K | Inlet temperature of HTF for dehydration and hydration |
| $q(x, \pm(1/2W_c + W_b), t) = q(L,y,t) = q(0,y,t) = 0$ | Adiabatic boundary |
| $u(0, -1/2W_c < y < 1/2W_c, t) = 25,30,35$ m/s | Velocity of HTF at entrance |
| At $y = \pm1/2W_c$ | No-slip of walls |
| $P_D(x,y,t=0) = 13{,}300$ P$_a$,$P_H(x,y,t=0) = 3000$ Pa | Initial partial pressure of steam for dehydration and hydration |
| $P_D(0, y, t) = 13{,}300$ Pa | Steam pressure at outlet for dehydration |
| $P_H(0, y, t) = 198{,}000$ Pa | Steam pressure at inlet for hydration |
| $P(x=L,y,t) = P(x,y = \pm1/2W_{c,t}) = P(x,y = \pm(W_b + 1/2W_c),t) = 0$ | No flux of steam |

## 3. Results and Discussion

### 3.1. Mesh and Numerical Model Analysis

The governing equations and boundary conditions were solved by the finite element method on the COMSOL Multiphysics. The absolute error was below $10^{-3}$, and $10^{-5}$ of turbulence variables and others respectively were considered to have met the condition of convergence. The separation solver was used to separate the variables of turbulence and others to increase the velocity of the solution and reduce memory usage although doing so sacrificed the rate of convergence. The turbulent kinetic energy and turbulence dissipation rate constituted a turbulent variable error group.

For the turbulent boundary layer, 35 layers of Y-Plus were used. We formed 30 layers of the turbulent boundary layer grid and 10 layers of the heat transfer boundary layer grid. The free triangular mesh was adopted in other areas. To ensure that the results were independent of grid quantity, three models were calculated based on grid density. n1 = 16,068, n2 = 55,588 for PFHS, n1 = 84,856, n2 = 201,680 for PPFHS and n1 = 183,558, n2 = 327,102 for HPPFHS (n1, n2 corresponding to two types of mesh number). The results of the conversion change in quicklime during the dehydration process under different grids are depicted in Figure 4. It can be seen that the higher grid density does not influence the results. Therefore, the mesh numbers of PFHS, PPFHS and HPPFHS models were 16068, 84856 and 183558, respectively.

In this work, except for the flow of heat transfer fluid, the reaction model was adopted based on the job of Qasim Ranjha et al. [12] which was verified by the relevant experimental data [21]. For the flow calculation of heat transfer fluid, the standard k–ε model was used instead of the laminar flow model. Using a standard k–ε model to study the strengthening of plate heat ex-changer by cylindrical flow under the air medium was very mature. The change in the calculation part of the heat transfer fluid flow will not affect the calculation of the rest of the physical field. As seen in Figure 4b, calculation results based on our model were highly consistent with that of Qasim et al. [12]. Steady flow was assumed in this study by ignoring the diffusion process of the high-temperature fluid at the beginning of the reaction process and the changing temperature gradient, which affects the fluid flow. As Figure 4 shows, because the temperature of the material bed did not exceed the reaction equilibrium temperature at beginning, so the reaction will proceed in the opposite direction in the mathematical model. This phenomenon was also reported in the relevant

references [12,14], and it did not affect the accuracy of simulation. In the beginning of the actual operation condition, this situation was reflected in the conversion rate of 0 [21].

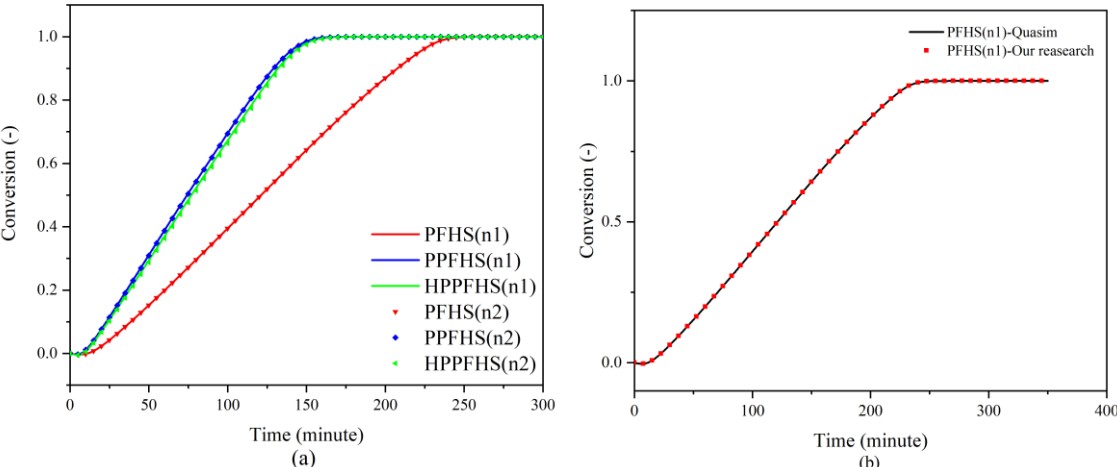

**Figure 4.** (**a**) Average conversion changes of the material over time during the dehydration by using PFHS (n1 & n2), PPFHS (n1 & n2) and HPPFHS (n1 & n2) reactors. (**b**) Average conversion changes over time by using PFHS (n1) model in this work and in Qasim's respectively. (dehydration, $\varepsilon$ = 0.5, u = 25 m/s).

### 3.2. Heat Transfer and Hydrodynamics Analysis

#### 3.2.1. Dehydration

Figure 5 shows the variation law of the mean temperature of the reaction porous bed during dehydration, and the whole reaction process can be divided into three stages. In the first stage, the whole system was set to 723.15 K, and the average temperature increased because of the heating of the HTF, which was 863.15 K at the entrance of the flow channel. The endothermic dehydration reaction was activated when the temperature of the porous bed exceeded the equilibrium temperature of the reaction. With the development of the endothermic dehydration reaction, the increase rate in the temperature of the reaction bed was slow in the second stage. In the third stage, the reaction rate decreased as the dehydration reaction deepened so that the heat absorption of the reaction gradually decreased, and eventually, the temperature of the bed stabilized.

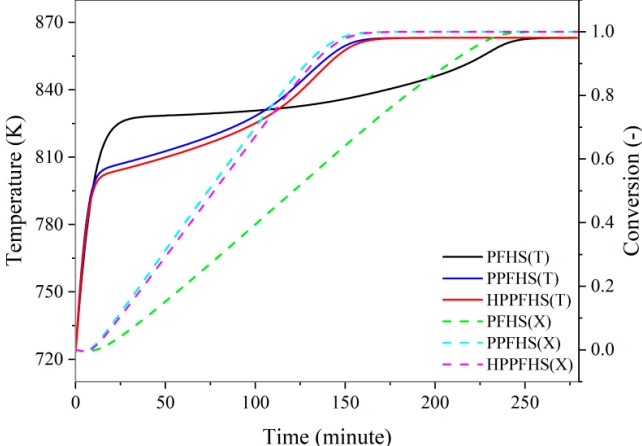

**Figure 5.** Average conversion (X) and temperature (T) of the porous bed over time for three heat transfer structures (dehydration, $\varepsilon$ = 0.5, u = 25 m/s).

It can be seen that the overall temperature trend was not affected by the differences in heat transfer structures (Figure 5). However, the mean temperature in the second stage of the reaction could be lower after adding fins because the endothermic reaction was

activated faster. As shown in the figure, the strengthening effects of PPFHS and HPPFHS structures were similar. To investigate the specific effect of the heat transfer structure in different reaction areas, two-point probes were applied at various distances from the flow channel with respect to PFHS and PPFHS according to geometric properties of the reactor. The coordinates of the points were listed in the Table 3.

**Table 3.** The coordinates of the six probe points in the porous bed.

| Name | Point.1 | Point.2 | Point.3 | Point.4 | Point.5 | Point.6 |
|---|---|---|---|---|---|---|
| X(mm) | 15 | 15 | 100 | 100 | 185 | 185 |
| Y(mm) | 5 | 10 | 5 | 10 | 5 | 10 |

As shown in Figure 6, the change law of conversion at different points in the bed was similar to that previously reported in the literature [14]. The conversion curves showed that the reaction proceeded faster near the channel after adding fins. For the conversion of P1 and P2, the time to complete the conversion was reduced by 39% and 36.8%, respectively. However, the time to achieve 50% conversion was reduced by 51.7% and 52.5%, respectively. Therefore, the degree of reinforcement was higher in the early stage of the reaction after adding fins. Although the reaction at P2 was enhanced more prominently than that at P1 in the first half-reaction process, the strengthening effect in P1 was better than that of P2 for the entire reaction process because the heat transfer performance of the area away from the HTF channel was lower due to the low heat conductivity coefficient along the bed. Thus, when the reaction process entered the second period, the temperature difference between the HTF and bed was lower than the first period, weakening the heat transfer effect. The detailed analysis of the heat transfer process was shown in the subsequent analysis of hydration.

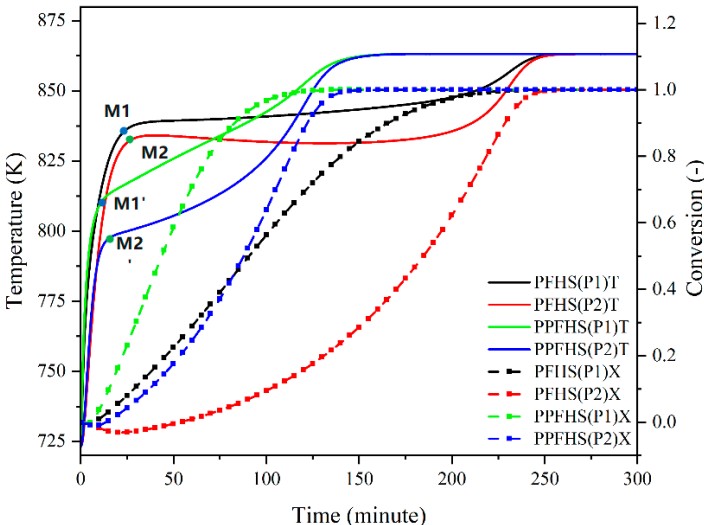

**Figure 6.** Temperature (T) and conversion (X) changed over time at the two-point probes (P1, P2) with respect to PFHS and PPFHS reactor (dehydration, $\varepsilon = 0.5$, u = 25 m/s).

Figure 6 shows that the average temperature of the porous bed was lower in the second stage after adding fins due to chemical reaction being excited more quickly, but the temperature trend was not changed at different points. We took the temperature inflection point (M) as an indicator. The temperature inflection points here meant that the heat transferred to the material by HTF is close to the heat absorbed by the material and the reaction began to move into the second phase. The temperature of inflection points reduced by around 30 K and 40 K for P1 and P2, respectively. Beyond that, the time of the inflection points was brought forward around 10 minutes after adding fins. All of the

above changes showed that reaction process was enhanced and the reaction went to the second stage earlier.

The fundamental cause of the changes in temperature and conversion curves is the enhancement of heat transfer. At the same time, the pressure drop of the fluid could be influenced due to the fins. Figure 7 shows the changes in the average heat flux along thin shells (Q) and pressure drop caused by the fins over time corresponding to the three types of reactors. The changing process of heat flux can be divided into three stages that correspond to different reaction stages. In the first stage, heat flux was used to heat the porous bed only and decreased enormously in a short time as the temperature difference between the fluid flow and shells decreased. Then, the endothermic reaction of the porous bed decreased the rising rate of the bed temperature so that the rate of the heat flux was lower in the second stage. Finally, as the reaction was complete, the bed temperature increased to that of the air, and the heat flux vanished. For different structures, the Q curve of the PFHS reactor in the early stage was lower than PPFHS and HPPFHS reactors and had a greater rate of decline. In the second stage, its trend was more stable, and it took longer for the heat flux to return to zero, indicating that the PFHS reactor had a longer reaction period than PPFHS and HPPFHS reactors. Regarding PPFHS and HPPFHS structures, it kept the same changing rule as Q curves of the PFHS reactor. Regarding air flow, the fluid disturbance in PPFHS and HPPFHS reactors occurred due to cylindrical fins and the motion pattern of fluid flow for PPFHS and HPPFHS is shown in Figure 8. The color represents the difference in the velocity of HTF. It indicated that the vortex was formed behind the cylinder because of fins, and the flow velocity increased after the fluid flowed through the fins in the mainstream area. The diameter of the channel at the location of fins was smaller, and the constant mass flow was the reasons for enhanced velocity. Leon et al. [24] investigated the influence of fin arrangement on convection and found that different arrangements lead to different pressure drops caused by fins. Yu et al. [16] and Yuan et al. [17] also found similar results. Figure 7 shows the pressure at the inlet of the flow channel for three kinds of reactors, indicating that the pressure drop of the fluid flow in the PPFHS and HPPFHS reactors was larger than that of the PFHS structure because additive fins impede fluid flow and form the vortex, which can be found more intuitively from Figure 8. The extra pressure caused by PPFHS was larger than that caused by HPPFHS, about 300 Pa. Under similar time to complete reactions, it could improve the operation cost.

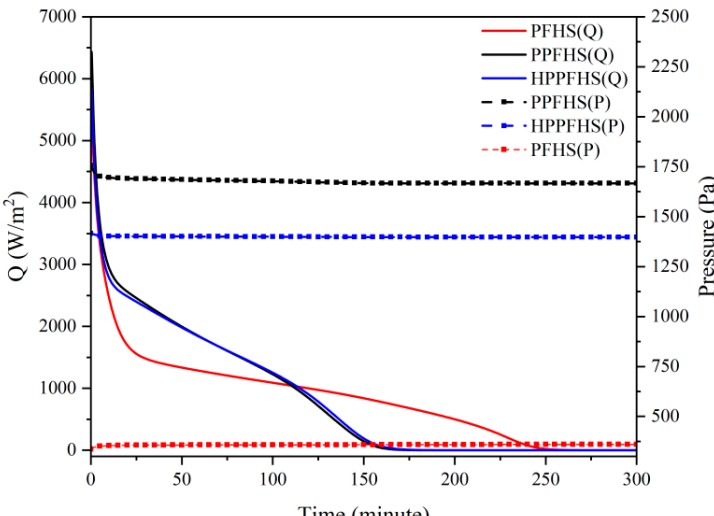

**Figure 7.** Heat flux (Q) of thin shells and pressure (P) at the inlet of the flow channel over time of three kinds of reactors (dehydration, ε = 0.5, u = 25 m/s).

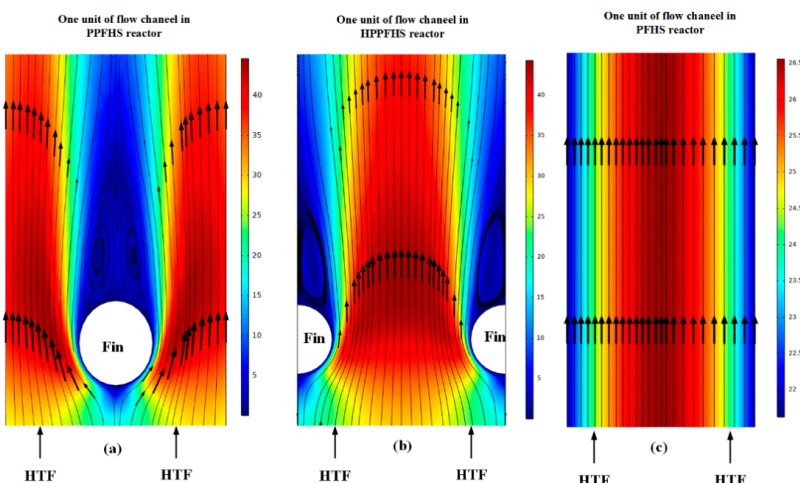

**Figure 8.** Streamline profile in the channel at 300 min (dehydration, $\varepsilon$ = 0.5, u = 25 m/s). (**a**) PPFHS reactor. (**b**) HPPFHS reactor. (**c**) PFHS reactor.

The velocity of HTF could influence the pressure drop of the fluid flow and convective heat transfer performance between the shells and HTF. In general, higher velocity means a higher convective heat transfer coefficient and pressure drop. Figure 9 depicted the average conversion changes and pressure drops over time at the entrance during dehydration at 25 m/s, 30 m/s and 35 m/s. It was shown that the increase in HTF velocity had limited effect on the reaction enhancement. However, the fluid pressure increased dramatically at the entrance with increasing the velocity of HTF. Therefore, exorbitant velocity of HTF was not necessary as it only has a small enhancing effect on the reaction process while an extra pressure drop would be caused and it would thus increase the operation cost.

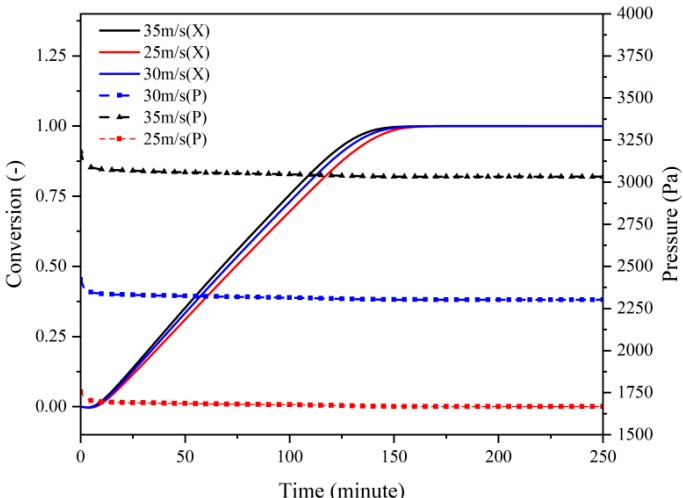

**Figure 9.** Conversion (X) curves and pressure (P) at the inlet of the flow channel of the PPFHS reactor under 25m/s, 30 m/s and 35 m/s. (dehydration, $\varepsilon$ = 0.5).

### 3.2.2. Hydration

Figure 10 indicates the average temperature and conversion rate of the bed changed over time. The hydration process could also be divided into three stages as the dehydration process. In the first stage, the initial temperature of the system and the temperature of HTF at the inlet of the flow channel was 623.15 K, which was lower than the reaction equilibrium temperature. Therefore, the hydration reaction was activated and the temperature of the bed increased rapidly. With the cooling effect of heat transfer between the fluids and bed, the temperature reached its peak and declined moderately in the second stage. Then, as

the reaction progressed, the exothermic reaction eventually stopped. The conversion kept constant, and the temperature went down to the same temperature as that of the HTF.

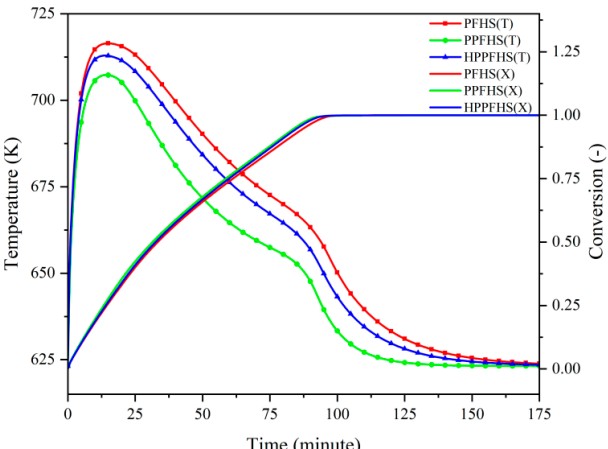

**Figure 10.** The average temperature (T) and conversion (X) of the material changed over time in the bed of three types of reactors (hydration, $\varepsilon = 0.5$, u = 25 m/s).

As seen in Figure 10, it is suggested that convective heat transfer enhanced by changing the reactor structure had little reinforcement on reaction process, but the temperatures of the porous beds in PPFHS and HPPFHS reactors were lower than that in PFHS reactor. The reason for this phenomenon was that heat source was the porous bed during hydration rather than HTF. Although the heat transfer performance was enhanced between shells and fluid flow (convective heat transfer), the thermal conductivity of the bed was still very low. Since convective heat transfer and heat conduction were series-wound, heat transferred from the bed to the wall was too low to affect the reaction process. In summary, heat conduction in the porous bed was the control step of the reaction process. Although the conversion of reactant could not be enhanced by improving convective heat transfer performance alone, lowering the temperature of the bed can decrease the heat loss of the porous bed during operation. PPFHS exhibited the best performance, the time to complete the reaction was reduced by 4%when compared with PFHS reactor.

In order to investigate the influence of thermal conductivity of the bed on the reaction process, the thermal conductivity of material system was assumed to be 2 W/m·K in the PFHS and PPFHS reactors. The corresponding samples were named PFHS (2 W/m·K) and PPFHS (2 W/m·K). Figure 11 indicates the conversion rate changed over time of them. It is shown that the time to complete reaction was reduced after improving thermal conductivity of the bed. However, enhancing convective heat transfer on this basis had no strengthening effect on the reaction process. Therefore, it was proven that the heat conduction along the bed dominates the overall heat transfer performance of the reactor and thus dominates the reaction process. Six different probe points were arranged in the bed as shown in Figure 3. Points 1, 3, and 5 were the probe points near the wall surface, and points 2, 4, and 6 were the probe points far from the wall surface. Figures 12 and 13 indicate the change in the conversion rate over time of probe points in three different reactors. Comparing PFHS with PFHS (2 W/m·K), it can be found that the reaction process was enhanced at points far from the wall. However, the time to completed reaction was improved at points near the wall, especially for the points 3 and 5. Therefore, we believed that improvement of heat conduction had a promoting effect on the conversion of material away from the heat exchange wall but had a negative effect on the conversion of material near the heat exchange wall. According to Figures 12b and 13b, the reaction process was enhanced near the wall side but had no effect far from the wall side. It could indicate that enhancing convective heat transfer performance had no effect on the reaction area away from the HTF which could prove that the low heat conduction performance in the bed had an adverse effect on the reaction process.

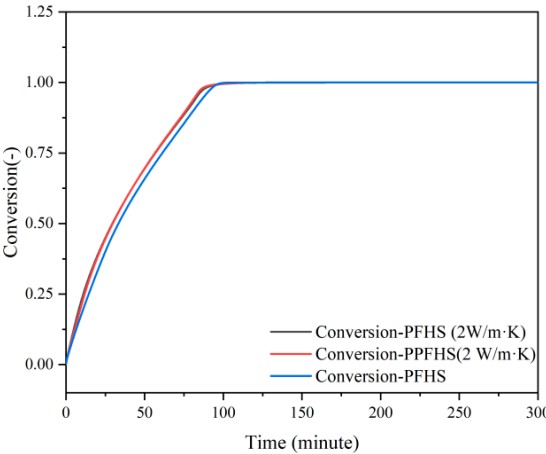

**Figure 11.** Average conversion rate (X) of the porous bed for PFHS, PFHS (2 W/m·K) and PPFHS (2 W/m·K) reactors (hydration, $\varepsilon = 0.5$, u = 25 m/s).

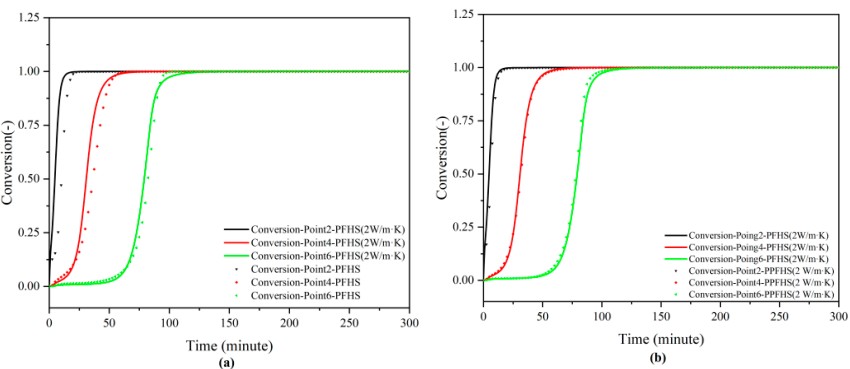

**Figure 12.** The conversion rate at the probe point far from the wall over time (hydration, $\varepsilon = 0.5$, 25 m/s). (**a**) PFHS (2 W/m·K) reactor compared with PFHS reactor. (**b**) PPFHS (2 W/m·K) reactor compared with PFHS (2 W/m·K) reactor.

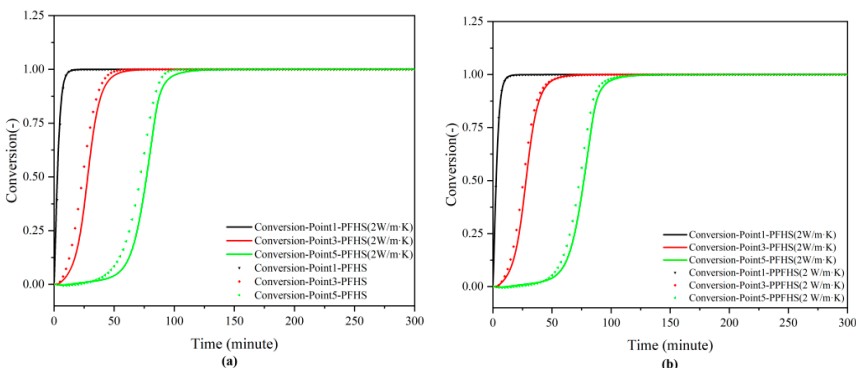

**Figure 13.** The conversion rate at the probe point near the wall over time (hydration, $\varepsilon = 0.5$, 25 m/s). (**a**) PFHS (2 W/m·K) reactor compared with PFHS reactor. (**b**) PPFHS (2 W/m·K) reactor compared with PFHS (2 W/m·K) reactor.

The reaction was triggered first and the heat released was transferred from the bed to the HTF during hydration. In this hydration process, heat conduction dominated the heat transfer process, which was different in the dehydration process under the same boundary conditions group. According to the analysis of conversion changed over time at different probe points, we found that the enhancement of heat conduction along the bed and convective heat transfer at the HTF side had different effects on the reaction process in different reaction areas. Improving the heat conduction of the bed material could enhance

the reaction process of material far from the HTF while improving the convective heat transfer could enhance the reaction process of material near the HTF.

### 3.2.3. Porosity

In this section, the influence of reactor structure on the reaction process at another bed porosity of 0.8 was further investigated according to the rules summarized above. Higher porosity would lead to lower thermal conductivity (~0.1 W/m·K) of the bed and charge amount of the reactor. Figure 14 presents the conversion and temperature changed over time for PFHS, PPFHS and HPPFHS reactors during the dehydration process. Comparing Figure 5 with Figure 14, it can be concluded that the modification of reactor structure had different effects on the reaction process under different porosity. PPFHS and HPPFHS reactors did not exhibit any reinforcement when the porosity was 0.8 compared with PFHS reactor. It is supposed that the excessively low thermal conductivity (~0.1 W/m·K) of material leaded to the difference between two kinds of porosity. Furthermore, the time to complete reaction was extended to around 250 minutes under porosity of 0.8, even though the amount of heat absorbed by the material was lower under this condition. Figure 15 shows the conversion and temperature of the bed changed over time during hydration process. It is found that the PPFHS reactor exhibited the best performance. In this case, the time to complete reaction reduced by 20 %. It has been reported that the time of completing the reaction was decreased from around 100 minutes to 30 minutes when the bed porosity increased from 0.5 to 0.8 [14]. The results in our work present a similar phenomenon.

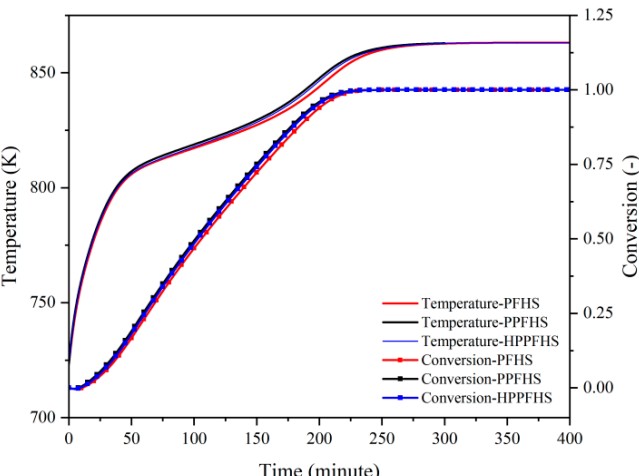

**Figure 14.** Conversion and temperature of the bed changed over time during dehydration ($\varepsilon = 0.8$, 25 m/s).

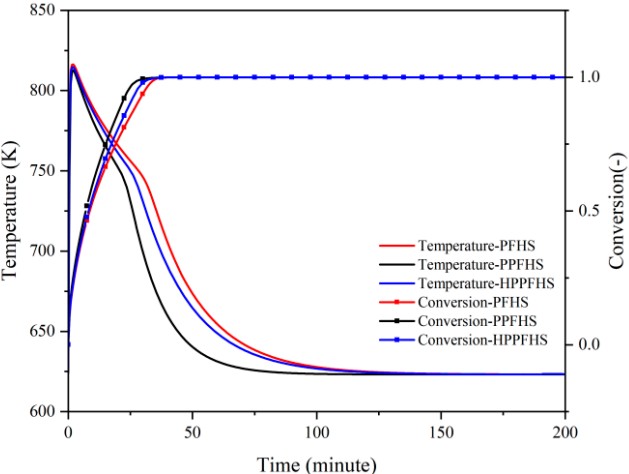

**Figure 15.** Conversion and temperature of the bed changed over time during hydration ($\varepsilon = 0.8$, 25 m/s).

By comparing the influences of reactor structure on the reaction process at different bed porosity, we found that the modified reactor structures had complicated effects on enhancing the reaction process. Under the condition of porosity being 0.5, modifying structure of reactor could obviously enhance dehydration, but had little effect on hydration. However, the modified structure had obvious reinforcement of reaction process on hydration but had no effect on dehydration when porosity was 0.8. We thought that two reasons lead to the difference in the effects of bed porosity on the hydration process. Firstly, when the porosity was 0.8, the total amount of material was less and the total amount of heat released was lower. Secondly, the reaction was more rapid, therefore the peak temperature was higher in the first reaction stage which made temperature difference between HTF and bed was bigger. Specifically, the second reason would directly result in the modified structure playing a role in the reinforcement of hydration reaction.

## 4. Conclusions

The heat transfer performance of reactors can be divided into convective heat transfer between the porous bed and fluid and heat conduction along the bed. As for different reaction process, they had different impacts on heat transfer performance under boundary conditions used in this paper. PPFHS and HPPFHS structures improved convective heat transfer performance by disturbing the fluid flow. Both structural reduced the time to complete reaction by about 33% during dehydration at the bed porosity of 0.5. As for the hydration process, they had little enhancement. After improving thermal conductivity of the bed, we obtained a noticeable improvement on conversion changed over time. Therefore, we thought that the heat conduction along the bed dominated the heat transfer process during hydration for a porosity of 0.5. The low thermal conductivity of material was the reason of this phenomenon. According to the analysis of conversion at six probe points, we found that the improvement of heat conduction could make the reaction finish early at points far from the heat exchanged wall, but the time to complete reaction was increased at points near the wall. Beyond that, enhancing convective heat transfer performance could bring opposite effect. Furthermore, both of them led to extra pressure drop, which increased the operation cost under the similar time to complete reaction. The extra pressure drops caused by PPFHS was larger than HPPFHS by about 300 Pa. Under the condition of a porosity of 0.8, the modified structure had no effect on the dehydration process due to the low thermal conductivity of the material. However, due to the higher temperature difference between HTF and the bed, the PPFHS reactor exhibited an obvious improving effect, which could reduce time to complete reaction by 20%. Excessive HTF velocity was unnecessary under the conditions in this paper as it would cause additional pressure drop along the HTF channel.

Above all, we propose that the HPPFHS had better performance under a porosity of 0.5, and the PPFHS reactor had better performance under a porosity of 0.8. When planning to enhance the reaction process by modifying the reactor structure, the heat conduction along the bed, convective heat transfer between HTF and the bed, and material parameters should be considered totally.

**Author Contributions:** Conceptualization, B.W. and Z.W.; methodology, B.W.; software, Z.W.; validation, B.W. and Z.W.; formal analysis, Y.L.; investigation, Y.M. and Y.L.; resources, Z.W.; data curation, B.W. and Y.M.; writing—original draft preparation, B.W.; writing—review and editing, B.W. and Z.W.; visualization, B.W.; supervision, Z.W.; project administration, Z.W.; All authors have read and agreed to the published version of the manuscript.

**Funding:** This research received no external funding.

**Data Availability Statement:** All the data used in this work have been shown in the manuscript.

**Acknowledgments:** Thanks to Cao Jun from East China University of Science and Technology for providing the help in using the software in this work.

**Conflicts of Interest:** The authors declare no conflict of interest.

**Nomenclature**

**Acronyms**

| | |
|---|---|
| PFHS | plate fin heat sinks |
| PPFHS | plate pin fin heat sinks |
| HPPFHS | half-plate pin fin heat sinks |
| PFHS (2 W/m·K) | plate fin heat sinks with extra heat conduction |
| PPFHS (2 W/m·K) | plate pin fin heat sinks with extra heat conduction |
| TCES | thermochemical energy storage |
| D | dehydration |
| H | hydration |
| HTF | heat transfer fluid |

**Symbol**

| | |
|---|---|
| A | Pre-exponential factor(1/s) |
| C | Specific heat, J/(kg·K) |
| E | Activation energy(J/mol) |
| $\Delta H$ | Enthalpy of reaction(J/mol) |
| K | Permeability |
| M | Molar mass(kg/mol) |
| n | Number of mesh elements |
| P | Pressure(Pa) |
| R | Rate of reaction |
| $S_Q$ | Heat source(W/m$^3$) |
| $S_m$ | Mass source(kg/(m$^3$ s)) |
| T | Temperature(K) |
| u | Velocity of HTF(m/s) |
| $V_{rs}$ | Molar density of solid reactant(mol/m$^3$) |
| $M_{st}$ | molar mass(kg/mol) |
| $W_b$ | Width of porous bed(mm) |
| $W_c$ | Width of flow channel(mm) |
| L | Length of porous bed(mm) |
| $\rho CaO$ | Density of CaO(g/cm$^3$) |
| $\rho Ca(OH)_2$ | Density of Ca(OH)$_2$,(g/cm$^3$) |
| $\lambda eff$ | Effective thermal conductivity of porous bed (W/m·K) |
| u | Velocity of air at entrance(m/s) |
| Por | Porosity of porous bed |
| r | Radius of cylindrical fins(mm) |
| dp | Diameter of grain(μm) |
| k | turbulent kinetic energy (m$^2$/s$^2$) |
| *E* | turbulent energy dissipation rate(m$^2$/s$^2$) |
| Q | heat flux between thin shells and HTF(W/m$^2$) |
| X | Conversion of solid reactant,1 |

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
