# Peer review of "Heat Transfer Enhancement of Indirect Heat Transfer Reactors for Ca(OH)2/CaO Thermochemical Energy Storage System"

_processes, doi:10.3390/pr9071136_

Round 1

Reviewer 1 Report

This paper presents an interesting analysis of lime thermal storage. Though the word lime is never used-I think you should at least mention that another name for CaO is quicklime and calcium hydroxide is also known as slaked lime. Overall, there needs to be much more explanation of the application, what the heat transfer fluid is, the parameters of the bed, and even whether the system makes thermodynamic sense with the large discrepancy between charging and discharging temperatures.

The typical temperature reaction should be specified in the introduction. Presumably it is quite high, so it might be something like liquid sodium coming from a concentrating solar power tower. You should explain where the steam goes that is driven off (unless it is captive). The size of the particles should be stated. You should say what the HTF is.

"Figure. 2. One research unit of the reactor. (a) Plate fin heat sinks (PFHS). (b) Half-plate pin fin heat sinks (HPPFHS) and (c) Plate pin fin heat sinks (PPFHS). "

(c) Difficult to see that the pins are not touching the walls (need to zoom in more). Seems like a strange choice because you're not increasing the heat transfer area at all. It would be very helpful to have an idea of the heat transfer coefficients on the two sides to see which one needs the surface area enhancement.

Figure 3 You should zoom in and use the same order as you did in figure 2.

PD(x,y,t = 0) = 13300 Pa,PH(x,y,t = 0) = 3000 Pa

Initial partial pressure of steam for dehydration and hydration

{These seem low - only 0.03 bar? Are there no other gases?}

Fig 6: conversion went negative?

Fig 8: specify that this is the flow of the HTF.

Fig 9: If the flow were turbulent, this should be 36% greater pressure drop for 35 m/s versus 30 m/s. However, you get about 110% more pressure drop. This could occur if you had a transition to turbulence, but I assume even a 30 m/s it would be turbulent (list the Reynolds number). So if you did not have a transition to turbulence, this is not reasonable.

Figure 10 caption needs to list where the temperature is measured.

Figure.12. The change of the conversion rate at the probe point far from the wall over time (hydration, ε=0.5, 25m/s). (a) PFHS(C) compared with PFHS(C). (b) PPFHS compared with PFHS(C).

{not consistent with figure. And what is (C)? Usually dots mean experimental data, but you just mean a different configuration, so I think it should just be a different type of line.}

It's not clear whether the steam is fed into and out of the system, or whether it is captive. If you have to feed the steam in and out, then you need a system to store the steam. So it seems like it would be much simpler just to store the heat transfer fluid. Then you would not have the massive hysteresis of charging versus discharging (inlet temps differ by 240 C, but I guess the outlet during hydration would be higher than the inlet – how much typically?), where you lose much of the ability of the heat to do work.

It's not clear whether porosity refers to inside the particles or of the entire bed. Again, a diagram would be very helpful of the bed.

A heatsink is when you transfer heat from a solid to a fluid. I guess that is appropriate with the solid CaO, but you also do have the steam fluid on that side.

You should say what the heat transfer coefficient is between the steam and the particles.

Reviewer 2 Report

The manuscript entitle "Heat transfer enhancement of indirect heat transfer reactors for 2 Ca(OH)2/CaO thermochemical energy storage system" shows a mathematical model to improve the efficiency of the thermochemical energy storage process by optimizing the structure of involved reactor.

This research sounds well, however, I would like to highlight the following points:

  1. In Introduction part, the goal of the work is not presented very well, which could be a problem for non-proficient readers.
  2. I encourage the authors to restructure the introduction section, in particular, what was commented in Figures 1-3, could be moved to Section 2 or 3.
  3. In Table 1 the structural and reaction parameters are reported based on previous works, however, it is not clear what is new in the present investigation, instead, only a few parameters are detailed through the text.
  4. As stated by the authors: "The specific mathematical model of the reactor was based on a previous study [13]". So what is the improvement with respect to the previous work (Ref 13.)
  5. Also , the boundary conditions are based on Ref. 15, and taking in mind point 3 and 4; a clear contribution of the present work is not observed. So, it might be interesting to propose novel boundary conditions.
  6. Although with the 2D model the computational cost is reduced, how is the value of 200 mm chosen? The authors have tested with different values of reactor length?
  7. Please double check the grammar/spelling here and there.

Reviewer 3 Report

The work concerns a very important technological issue. The research problem has been clearly presented. The work requires minor adjustments.

1. Please complete the affiliation of the authors of the article
2. Please complete the email addresses of the authors of the article so that they refer to the institution they represent
3. Figure 13 - please pay attention that the charts have the same and good, legible quality

Round 2

Reviewer 1 Report

I see that it would not be feasible to store hot air. It still might make sense to store steam, however, as you list that is one of the heat transfer fluids. Overall, you have addressed my concerns. However, your new text has a lot of typos that need to be corrected.

Author Response

Grammar and spelling have been further examined in our manuscript.

Thank you for your efforts in this work.

Best wishes.

Reviewer 2 Report

That authors have considered in detail some commented points. I recommend the current version for publication.

Author Response

Spelling and grammar have been further checked in our manuscript.

Thank you for your efforts in this work.

Best wishes.